# Single-cell lineage tracing by endogenous mutations enriched in transposase accessible mitochondrial DNA

Jin Xu[1,2,3], Kevin Nuno[4,5], Ulrike M Litzenburger[1,2,3], Yanyan Qi[1,2,3], M Ryan Corces[1,2,3], Ravindra Majeti[4,5]*, Howard Y Chang[1,2,3,6]*

[1]Center for Personal Dynamic Regulomes, Stanford, United States; [2]Department of Dermatology, Stanford University School of Medicine, Stanford, United States; [3]Department of Genetics, Stanford University School of Medicine, Stanford, United States; [4]Institute for Stem Cell Biology and Regenerative Medicine, Stanford University School of Medicine, Stanford, United States; [5]Division of Hematology, Department of Medicine, Stanford University School of Medicine, Stanford, United States; [6]Howard Hughes Medical Institute, Stanford University, Stanford, United States

**Abstract** Simultaneous measurement of cell lineage and cell fates is a longstanding goal in biomedicine. Here we describe EMBLEM, a strategy to track cell lineage using endogenous mitochondrial DNA variants in ATAC-seq data. We show that somatic mutations in mitochondrial DNA can reconstruct cell lineage relationships at single cell resolution with high sensitivity and specificity. Using EMBLEM, we define the genetic and epigenomic clonal evolution of hematopoietic stem cells and their progenies in patients with acute myeloid leukemia. EMBLEM extends lineage tracing to any eukaryotic organism without genetic engineering.
DOI: https://doi.org/10.7554/eLife.45105.001

*For correspondence:
rmajeti@stanford.edu (RM);
howchang@stanford.edu (HYC)

## Introduction

Resolving lineage relationships between cells is necessary to understand the fundamental mechanisms underlying normal development and the progression of disease. In recent years, new methods have emerged to enable cell lineage tracking with increasing resolution, leading to substantial biological insights (*Woodworth et al., 2017*). Specifically, genome editing of reporter constructs via CRISPR-Cas9 allowed synthetic reconstruction of cell lineage relationships in model organisms, and has been coupled with transcriptome profiling to inform cell fates (*Spanjaard et al., 2018*). These prospective 'mutate-and-record' methods provide powerful tools to resolve the developmental origin of cells in genetically engineered cells and organisms, but cannot be utilized in living humans, archival clinical samples, or any wild type organism (*Woodworth et al., 2017*). Given these limitations, retrospective lineage tracing using endogenous genetic markers is an alternative solution. Recent advances in sequencing enable naturally occurring somatic mutations to be used as lineage markers, which usually required single-cell genome sequencing to capture the sparse genetic information (*Lee-Six et al., 2018*; *Wang et al., 2014*). Regions with high mutation rates, such as microsatellite repeats, retrotransposons, and copy-number variants, has been used to resolve the lineage relationship for normal or cancerous tissue samples (*Evrony et al., 2015*; *Biezuner et al., 2016*). These methods reduce the cost of whole genome sequencing, but still lack information on cell phenotypes.

Simultaneous measurement of the lineage relationship and cell fates is ultimately required to address many biomedical questions. Here we describe EMBLEM (<u>E</u>pigenome and <u>M</u>itochondrial

Barcode of Lineage from Endogenous Mutations), a strategy to track cell lineage using endogenous mitochondrial DNA variants in ATAC-seq data. The end result of EMBLEM is single-cell lineage information and rich global epigenomic profile from the same individual cells (*Figure 1A* and *Figure 1—figure supplement 1*).

We illustrate the utility of EMBLEM in human blood progenitor cells to clarify the process of pre-leukemic clonal evolution and the emerging biology of clonal hematopoiesis.

## Results

Assay of Transposase-Accessible Chromatin by sequencing (ATAC-seq) is a sensitive method used to study chromatin accessibility profiles in diverse cell types and organisms (*Buenrostro et al., 2013*). During DNA transposition and amplification in cells, mitochondrial DNA is also amplified at the same time (*Figure 1A*). Mitochondrial DNA (mtDNA) is a ~ 16 kb circular genome with ~10 fold higher mutation rate compared to the nuclear genome. Hence, mtDNA incrementally accumulates unique, irreversible genetic mutations that are passed on to daughter cells even in healthy humans and may be used for lineage tracing (*Morris et al., 2017*), (*Fellous et al., 2009*). The majority of somatic mtDNA mutations are noncoding and thought to be passenger (*Ju et al., 2014*). Importantly, the number of mitochondria (and therefore mtDNA) range from several hundreds to >10,000 per cell in different cell types, facilitating robust mtDNA analysis even from a single cell.

We first observed that ATAC-seq effectively enriches for mtDNA. While mtDNA is present in many kinds of DNA sequence libraries, it is substantially enriched in ATAC-seq libraries due to the fact that mtDNA is not chromatinized and is therefore highly accessible (*Supplementary file 1*). ATAC-seq enables a 17-fold or greater enrichment of mtDNA compared to exome sequencing or whole genome sequencing in GM12878 human B cells (*Figure 1B*), leading to an average ~18,000X coverage of mtDNA (*Figure 1—figure supplement 2A*). With this coverage, we detected 27 mitochondrial variants from GM12878 cells (*Figure 1C*). 13 of these variants have a variant allele frequency (VAF) greater than 90%, which are known as homoplasmic variants (*Figure 1—figure supplement 2B*). We also detected 14 low frequency mitochondrial DNA variants, with VAFs ranging from 0.1% to 24% (*Figure 1C* and *Figure 1—figure supplement 2C*). Similar results for mtDNA enrichment were observed in human K562 cells (*Figure 1—figure supplement 3*, *Supplementary file 1*).

The VAF from bulk ATAC-seq data represents the average of the allele frequencies of the cell population. A 25% VAF may arise from 25% of cells in the population with a homoplasmic variant, or alternatively arise from 100% of cells all having a quarter of their mitochondria with the variant allele (*Figure 1D*). To distinguish between these two models, we analyzed single-cell ATAC-seq data from GM12878. For four mtDNA variants (VAF between 0.5% ~ 24% at population level), we find that a mixture of both models is in action for different variants (*Figure 1E*). For instance, mtDNA mutation 3082 is widely spread among single cells, but at low frequency per cell. Because it is extremely unlikely (see Materials and method) that the identical mutation arose independently in every single cell, cells sharing the same mitochondrial mutations are inferred to have descended from the same ancestral cell. These results suggest that even low frequency heteroplasmic mtDNA mutations can be exploited for lineage tracing.

To prove the principle that somatic mitochondrial mutations can track cells from the same ancestor and to quantify the power of lineage mapping, we next applied EMBLEM to primary blood cells from patients with acute myeloid leukemia (AML). Human AML is organized as a hierarchy: a hematopoietic stem cell first acquires an initiating mutation in one of a number of chromatin modifier genes, previously termed as 'pre-leukemic' hematopoietic stem cell (pHSC) (*Jan et al., 2012*; *Corces-Zimmerman et al., 2014*). pHSCs are functionally normal and are not able to transplant AML, but upon accumulation of additional mutations, they give rise to leukemic stem cells (LSCs) that are able to self-renew and recapitulate AML disease upon transplantation (*Jan et al., 2012*; *Thomas and Majeti, 2017*). Finally, LSCs give rise to the bulk leukemic blast cells in AML (*Thomas and Majeti, 2017*). Targeted exome sequencing in these samples have identified somatic mutations in tumor suppressor genes and oncogenes that link the lineage relationship of pHSCs, LSCs and blasts, providing the ground truth for our analyses (*Corces-Zimmerman et al., 2014*).

We applied EMBLEM to the ATAC-seq profiles of FACS-purified LSCs and leukemic blasts first (*Supplementary file 1*). Using high-confidence mtDNA mutations, detected both from bulk ATAC-

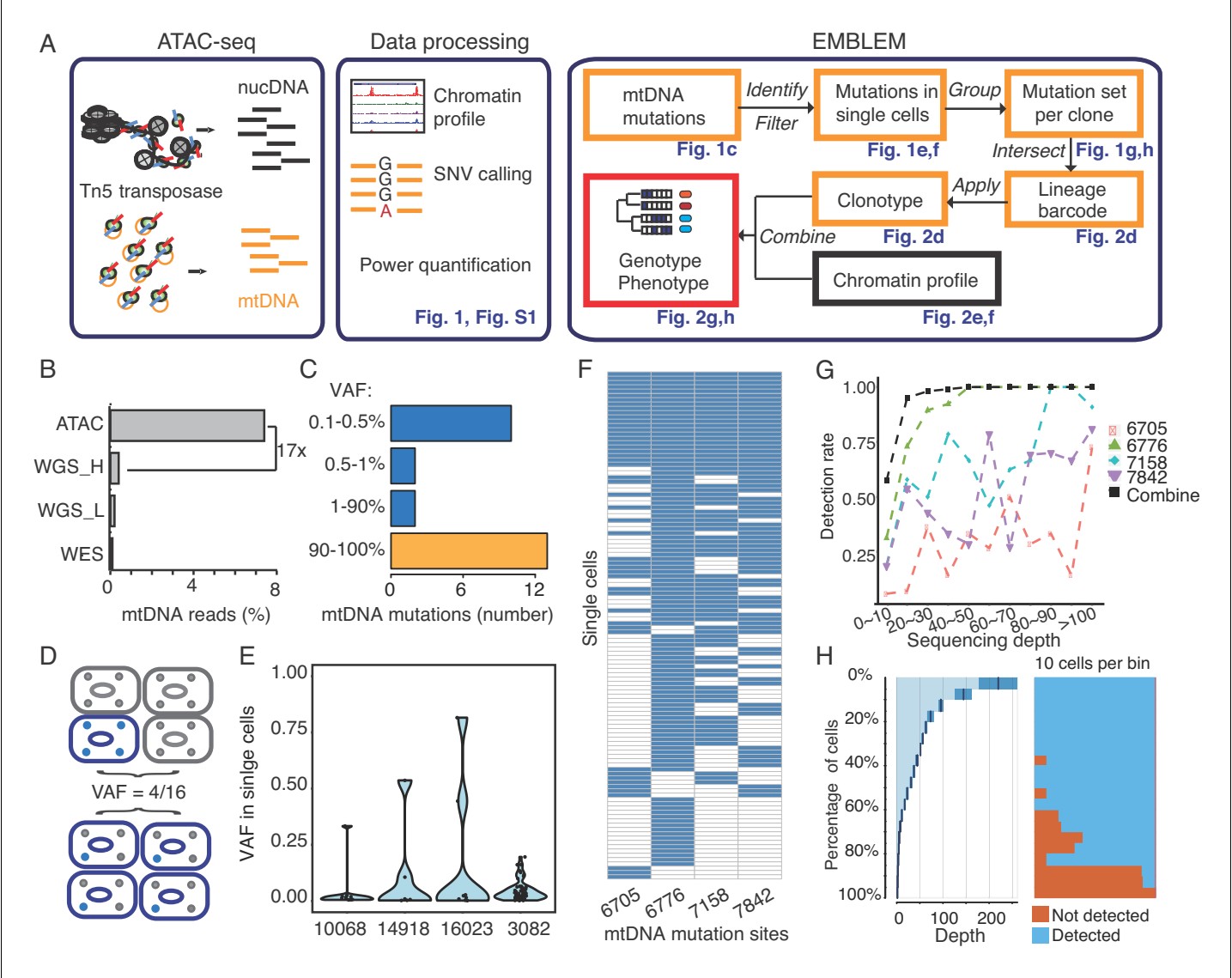

**Figure 1.** EMBLEM reveals cell lineage from mtDNA mutations. (**A**) EMBLEM workflow. Usings standard ATAC-seq data as input (left), an SNV calling step was added to enumerate all single nucleotide variants in mtDNA (middle). EMBLEM identifies heteroplasmic mtDNA mutations in single cells, groups mutations into diagnostic sets, and infers cell lineage based on mtDNA variants, and overlays clonotype information on epigenomic profile of the same cells (right). (**B**) ATAC-seq enriches for mtDNA reads compared to whole exome sequencing (WES), low coverage whole genome sequence (WGS_L), or PCR-free, high-coverage whole genome sequence (WGS_H). (**C**) Bimodal distribution of variant allele frequency (VAF) of mtDNA mutations discovered using ATAC-seq. Yellow bar presents the homoplastic variants that can distinguish different individuals. Heteroplasmic variants can distinguish clonal cell populations within one individual. (**D**) Two possible models for 25% mtDNA VAF in bulk: Homoplastic variants in a small proportion of cells (top) or heteroplasmic variants in nearly every single cell (bottom). Blue cells: cells with mutated mtDNA, blue dots: mtDNA with mutated allele. (**E**) VAF of mtDNA mutations in single cell ATAC-seq data of human B cells. Each dot presents the VAF (y-axis) in single cells, and rotated kernel density on each side presents their distribution. The x-axis indicates the mutation site (the nucleotide position in mitochondrial genome). (**F**) mtDNA mutations in human AML. Each row in the heap map is a single cell (LSC or AML blast); each column is a heteroplasmic mtDNA mutation. Blue color indicates the mtDNA variant is detected (>1 reads); white color indicates no mutation. The nucleotide position in mitochondrial genome for each mutation is indicated. (**G**) Combined set of heteroplasmic mtDNA mutations improve cell lineage assignment in single cells. Cells were first separated into bins according to their mtDNA coverage (x-axis). The detection rate (y-axis) for each site (indicated by different color and shape) is calculated with the number of cells with that mutations divided by total number of cells in that bin. The detection rate of combining four sites (black line, METHOD) is substantially increased. (**H**) Quantitation of mtDNA mutation detection rate as a function of sequencing depth and number of single cells. Cells were sorted in descending order by their sequencing depth and grouped into bins (10% of cells in each row). Distribution of sequencing depth is shown on the left panel. The black line and dark blue shade indicate mean ± standard deviation, respectively. The light blue shade indicates remaining value of the bin. Cells with or without mtDNA variants are shown in blue and orange on the right panel, respectively.

DOI: https://doi.org/10.7554/eLife.45105.002

*Figure 1 continued on next page*

*Figure 1 continued*

The following figure supplements are available for figure 1:

**Figure supplement 1.** EMBLEM workflow for SNP calling and lineage inference.

DOI: https://doi.org/10.7554/eLife.45105.003

**Figure supplement 2.** mtDNA coverage and variants from different sequencing libraries from GM12878 human B cells.

DOI: https://doi.org/10.7554/eLife.45105.004

**Figure supplement 3.** Heteroplasmic mtDNA mutation in K562 cells.

DOI: https://doi.org/10.7554/eLife.45105.005

**Figure supplement 4.** Heteroplasmic variants in single cells from AML blasts and LSCs (SU353).

DOI: https://doi.org/10.7554/eLife.45105.006

**Figure supplement 5.** Heteroplasmic variants in single cells from AML blasts and LSCs (SU070).

DOI: https://doi.org/10.7554/eLife.45105.007

seq and single cell ATAC-seq, we found the LSC and blast populations not only shared the same heteroplasmic variants, but also showed similar distribution and allele frequency at the cellular level (*Figure 1—figure supplement 4*). These results indicate the two populations are identical at the genetic level, but divergent at the epigenomic level, consistent with previous studies (*Corces et al., 2016*; *Schep et al., 2017*). In patient SU353, we identified four diagnostic mtDNA mutations in the same cell (*Figure 1F*), which indicates these four mitochondrial variants already co-existed in the ancestral cell (see Materials and method). With the assumption that all these LSCs and blasts are clonal, we further quantified the detection rate of each mtDNA variant as a function of allele frequency and sequencing depth (*Figure 1G*). We found that when a single variant allele has a frequency greater than 20%, the detection rate can be up to 90% with >20X coverage (e.g. site 6776). In contrast, when the variant allele has a frequency lower than 1%, the detection rate drops to 20% when the coverage is below 100X (e.g. site 6705). While high drop-out rate is a common challenge for single-cell technologies (*Kharchenko et al., 2014*), computational imputation of the missing information from single cell data can address this problem (*Zhang and Zhang, 2018*). When multiple mtDNA variants are co-detected in multiple single cells, we can infer their origin and linkage in the ancestral cell (see Materials and method). Thus, cells containing any one of these variants will still inform their origin from the same lineage. With any combination of the four variants, 90% (sensitivity) of the cells can be unambiguously assigned to the correct lineage with just 20x mtDNA coverage (*Figure 1H*). Furthermore, two mtDNA mutations identified in other cells (e.g. pHSC specific site 2967,6268) were never detected (false positive = 0) in LSCs and blasts (*Figure 1—figure supplement 4*), showing a high specificity of the method. Similar performance of single cell lineage tracing for another patient (SU070) is shown in *Figure 1—figure supplement 5*. These results demonstrate that somatic DNA mutations in the mitochondrial genome are a powerful endogenous marker to identify clonal cell populations.

To expand on these findings to additional different cell lineages, we applied EMBLEM to bulk ATAC-seq data from sorted blood cells from healthy human donors and patients with AML (*Supplementary file 1*) (*Corces et al., 2016*). We identified heteroplasmic mtDNA mutations in multiple cell populations of primary blood cells from healthy donors and all AML patients (*Figure 2—figure supplement 1A*, *Supplementary file 2*). The heteroplasmic mtDNA mutations showed a similar mutant spectrum as observed by previous studies using cancer genomic data (*Figure 2—figure supplement 1B and C*) (*Ju et al., 2014*).

Furthermore, EMBLEM, not only confirmed the previous lineage hierarchy of AML, but also extended the previous model of pHSC heterogeneity (*Figure 2A*). In the AML cases with LSCs sequenced by ATAC-seq, the LSCs and their corresponding leukemic blasts have nearly identical heteroplasmic mtDNA mutations (*Figure 2B–C* and *Figure 2—figure supplement 1D*), suggesting a direct lineage relationship and short generation history between LSCs and blasts. We then examined whether any of the mtDNA variants present in LSCs can be seen in the pHSCs, where the first leukemia-associated protein-coding mutations have already occurred in functional normal hematopoietic stem cells (*Corces-Zimmerman et al., 2014*; *Corces et al., 2016*). We detected blast-associated mtDNA mutations in pHSCs in all 11 cases. Interestingly, we also detected additional heteroplasmic mtDNA mutations present specifically in pHSCs (*Figure 2C*). In the 11 cases we investigated, seven cases have pHSC-unique heteroplasmic mtDNA mutations (*Figure 2—figure supplement 1D and*

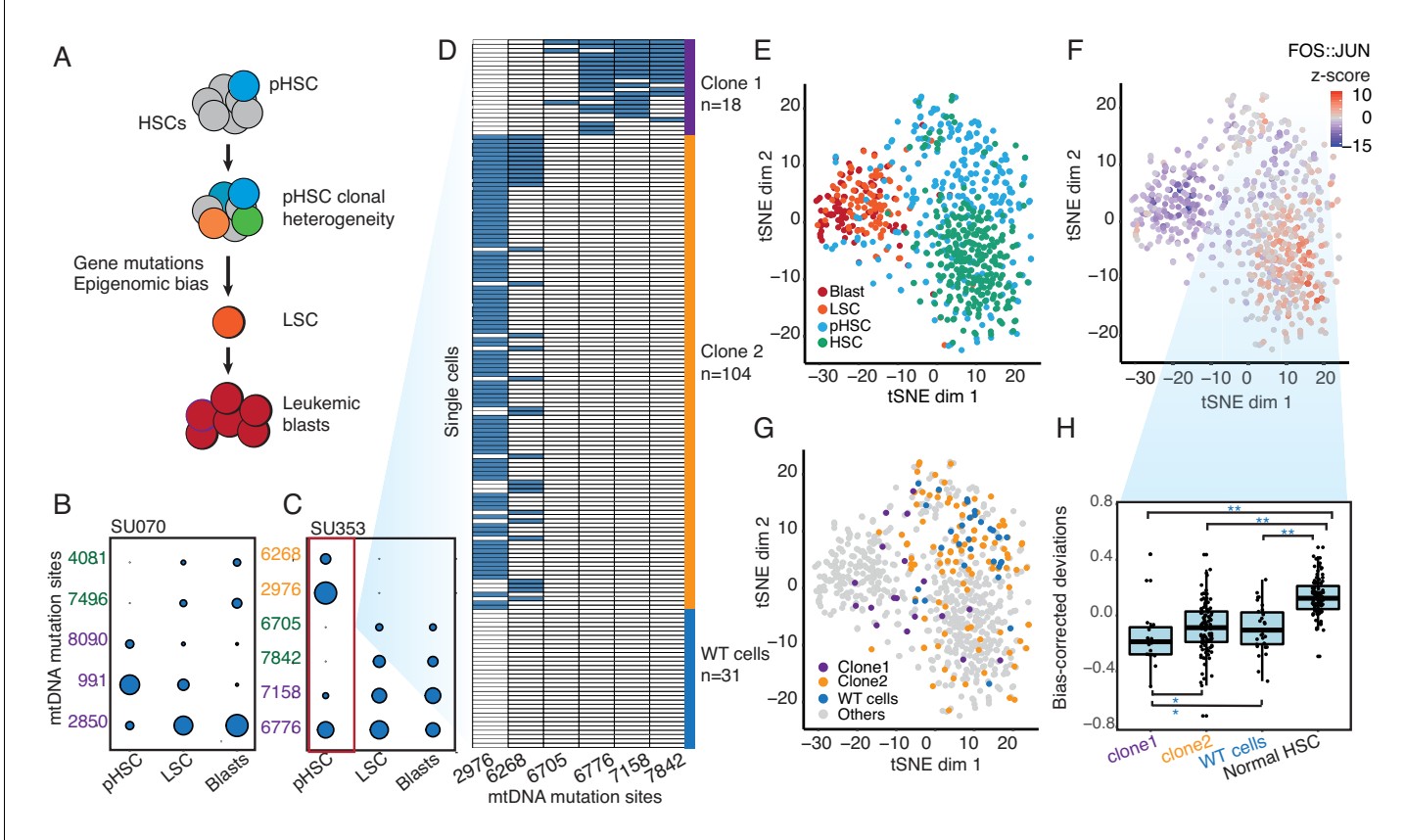

**Figure 2.** Clonal evolution of pre-leukemic HSCs inferred from joint lineage tracing and single cell chromatin accessibility. (**A**) Lineage hierarchy in acute myeloid leukemia based on EMBLEM and prior genetic information. mtDNA mutations reveals pHSC clonal heterogeneity. The clonal precursor of the leukemic stem cell is not the clone with most representation in the pHSC pool, but rather the clone with epigenomic bias towards the leukemic regulatory program, as depicted by related color schemes. (**B**) EMBLEM deconvolutes AML clonal heterogeneity. Heteroplasmic mtDNA mutations in three cell populations from patients SU070 are shown. Mutations sites (in rows) in each FACS-sorted cell population (in columns) are shown, with size of each circle representing its VAF. Several mtDNA mutations (sites shown in purple) are detected in pHSCs and transmitted to LSCs and blasts, confirming those pHSC clones at the apex of leukemia lineage. LSCs accumulated additional mtDNA mutations (sites shown in green) and are transmitted to leukemic blasts in patient SU070. Allele frequency, sequencing depth and annotation of the variant allele are shown in *Figure 2—figure supplement 1* and *Supplementary file 2*. (**C**) Same plot as (**B**) shown for patient SU353. In addition to the shared mtDNA mutations in pHSCs, LSCs, and blasts (purple), two pHSCs-specific mtDNA mutations are also detected (yellow). Allele frequency, sequencing depth and annotation of the variant allele are shown in *Figure 2—figure supplement 1* and *Supplementary file 2*. (**D**) Heteroplasmic mutations in single pHSCs from one patient reveal clonal heterogeneity. Each column is a mtDNA nucleotide position; each row is one cell. Blue color indicates the presence of the mtDNA variant. Shown are cells with any mtDNA mutation detected, or cells with more than 40X coverage of the mitochondrial genome without any detected mutation (pHSC with WT mtDNA). The number of cells in each clonotype are indicated on the right. (**E**) Landscape of single-cell chromatin accessibility of blood progenitor and leukemic cells in patient SU353. tSNE map using bias-corrected deviations from chromatin accessibility showing cluster of AML blasts, LSCs, pHSCs and normal HSC, colored by cell types. (**F**) Chromatin accessibility of the FOS:JUN binding motif across the same single cells. tSNE map colored by deviation z-score for motif associated to FOS:JUN, the most variable TF motif. (**G**) pHSC clones possess distinct epigenomic signatures. Clone 1 that gives rise to the AML has a chromatin accessibility profile that more resembles LSCs and leukemic blasts. 'WT' pHSC refers to the pHSC with WT mtDNA. Clonotype information from EMBLEM is overlaid on the tSNE map defined by TF motif deviations, and colored by different lineal sub-populations defined by mtDNA mutations. (**H**) Quantitation of distinct single-cell chromatin accessibility at FOS:JUN motifs among different pHSC clones defined by EMBLE. Clone 1 pHSCs tend to down regulate FOS:JUN accessibility, while clone 2 pHSC shows substantially greater cell-to-cell variability. pHSCs with no detectable mtDNA variants and normal HSCs are shown for comparison. TF deviation of single cells (black dots) is shown on the distribution box-plot. The statistical significant were indicated by '*' when p<0.05, '**' when p<0.01(Wilcoxon rank-sum test).

DOI: https://doi.org/10.7554/eLife.45105.008

The following figure supplements are available for figure 2:

**Figure supplement 1.** Heteroplasmic mtDNA mutations detected in bulk ATAC-seq from AML patients.

DOI: https://doi.org/10.7554/eLife.45105.009

**Figure supplement 2.** Single cell chromatin accessibility.

DOI: https://doi.org/10.7554/eLife.45105.010

*Figure 2 continued on next page*

*Figure 2 continued*

**Figure supplement 3.** Sorting Scheme for pHSCs.
DOI: https://doi.org/10.7554/eLife.45105.011
**Figure supplement 4.** Investigation of horizontal mitochondrial transfer using mixing experiment from mouse and human cells.
DOI: https://doi.org/10.7554/eLife.45105.012

*E*), a previously unrecognized level of pHSC heterogeneity. pHSCs are capable of long-term self-renewal and possess a clonal growth advantage, allowing them to clonally outcompete normal HSCs. Indeed, the clonal frequency of pHSCs is a poor prognostic factor for overall survival in AML (*Corces et al., 2016*). Our discovery of pHSCs with distinct heteroplasmic mtDNA mutations suggests the existence of multiple distinct sub-clones of pHSCs in AML patients.

To validate the heterogeneity of pHSCs inferred from EMBLEM of bulk cell populations, we performed single-cell ATAC-seq of HSCs from AML patient SU353, which exhibited both a high burden of pre-leukemic somatic coding gene mutations and high frequency of pHSC-specific heteroplasmic mtDNA mutations (*Corces et al., 2016*). We identified the heteroplasmic mtDNA variants from each single cell, which separated the HSCs into three lineages: Two clonal subpopulations termed 'clone 1' (18 cells) and 'clone 2' (104 cells), and a third population with no mtDNA variants despite sufficient mtDNA coverage (pHSC with WT mtDNA, 31 cells) (*Figure 2D*). Notably, clone 2 possessed pHSC-specific mtDNA mutations, while clone 1 possessed mtDNA mutations shared with LSCs, indicating clone1 is the lineage precursor of AML. These results confirm that multiple pHSC clones arise in AML patients, and one subclone eventually evolved to become the LSC (*Figure 2—figure supplement 2A*).

Finally, we related the clonotype of pHSCs to their single-cell chromatin accessibility profiles. We interrogated the patterns of active DNA elements and enriched transcription factor motifs in sequential stages of AML development from the same patient, and contrasted with HSCs from normal donors using ChromVAR (*Schep et al., 2017*) (*Figure 2E* and *Figure 2—figure supplement 2B*). The chromatin accessibility profiles of pHSCs are more similar to HSCs than to LSCs or leukemic blasts. The greatest deviation between HSC and other cell types occurred at DNA binding motifs of the transcription factor Jun/Fos, a known key regulator of HSC biology (*Santaguida et al., 2009*) (*Figure 2F*). Furthermore, the three lineages of pHSCs revealed by mtDNA mutations also showed distinctive chromatin profiles (*Figure 2G*). Clone 1 pHSC, which gives rise to the LSC and AML leukemia, is already more similar to LSCs and blasts in its chromatin accessibility. In contrast, clone 2 that comprises the larger fraction of pHSCs exhibited variable chromatin profiles at the single-cell level that spanned the range of normal HSCs, pHSC with WT mtDNA(WT cells), is also diverged from normal HSCs. Thus, both lineage tracing and single cell epigenomic states indicate clone 1 as the original stem cell of the AML in patient SU353. Supervised comparison of the chromatin accessibility profiles among these clonal sub-populations further identified distinct and significantly enriched transcription factor motifs (*Figure 2H* and *Figure 2—figure supplement 2C–E*). These results indicate the heterogeneity of HSCs from AML patients both on a genetic and epigenomic level.

## Discussion

We present a computational strategy to combine cell lineages tracing by endogenous mtDNA mutations and chromatin accessibility profiling in the same cell using single-cell ATAC-seq data. This approach is applicable to any eukaryote, does not require genetic engineering or genome editing, and is cost effective as the lineage information comes 'for free' on top of epigenomic insights. The relative merits of mtDNA vs other genetic markers for lineage tracing are outlined in *Supplementary file 3*. An important advantage of EMBLEM is that we enable clonotype tracing in existing ATAC-seq data sets and hierarchical lineage construction from ATAC-seq that thousands of labs have already generated. All future ATAC-seq data acquired for other inquiries will also have the benefit of lineage information. EMBLEM may also be extended to other single cell technologies, in which mtDNA is sequenced. We show that EMBLEM is successful even with low frequency heteroplasmic mutations, detection of rare clones in a population, and authentic clinical samples. With advances in the throughput and depth of single-cell genomic technologies, we believe EMBLEM may be a powerful tool to bring insight for many biomedical questions, including development,

regeneration, immunity, and cancer with integration of genotype and phenotype information from the same cell. During revision of this work, *Ludwig et al. (2019)* reported the feasibility of using mtDNA and single cell genomics for lineage tracing, which independently validates the potentially broad utility of this approach.

Although powerful and broadly applicable, mtDNA lineage tracing also has its limitations. One limitation of this method is absence of mtDNA mutations in cells and tissues of embryos and young animals, which precluded us from applying EMBLEM to published scATAC-seq data of early animal development. Moreover, the possibility of selective mitochondrial inheritance or intercellular mitochondria transfer may affect the accuracy of inferred lineages (*Mishra and Chan, 2014*; *Moschoi et al., 2016*; *Marlein et al., 2017*; *Hayakawa et al., 2016*). On the other hand, asymmetric transmission of mitochondria would not necessarily affect cellular lineage tracing, as long as the variant alleles are randomly segregated. Using scATAC-seq data from a mixing experiment with human and mouse cells (*Satpathy et al., 2018*), we found species-specific mtDNA always paired with species-specific nuclear genomic DNA (*Figure 2—figure supplement 4*). These results suggest that mitochondrial horizontal transfer is not a confounder of our study and does not universally occur between cells. The aforementioned two scenarios reflect the potential uncoupling of nuclear and mitochondrial genomes, which would be of interest to investigate by EMBLEM in combination with other nucDNA tracing methods.

mtDNA lineage tracing produced new insights concerning the pHSC, the human hematopoietic stem cell that suffers the first oncogenic mutation in AML evolution. Our results add to the evidence that the pHSC population is heterogeneous, with evidence of multiple mtDNA clones. Unexpectedly, the pHSC lineage that gives rise to the subsequent acute myeloid leukemia is not the lineage with the best competitive potential among pHSCs, as the leukemogenic lineage is often in the minority. pHSC burden is a strong poor prognostic predictor of AML survival (*Corces et al., 2016*). It is widely believed that the association between high pHSC burden and poor AML patient prognosis reflected the enhanced self-renewal and competitive ability of the mutant pHSC. Our analysis suggests that high pHSC burden may reflect the diversity of pHSCs or the underlying mutational processes. These alternative interpretations of the link between pHSC burden and poor clinical prognosis should be addressed in future studies.

## Materials and methods

### Public data accession

Aligned bam files for GM12878 whole exome, low coverage whole genome, and PCR free whole genome sequence, were downloaded through phase 3 release of 1000 genomes (ftp://ftp.1000genomes.ebi.ac.uk).

The alignment files were accessed via the following ftp links:

- ftp://ftp.1000genomes.ebi.ac.uk/vol1/ftp/phase3/data/NA12878/exome_alignment/NA12878.mapped.ILLUMINA.bwa.CEU.exome.20121211.bam
- ftp://ftp.1000genomes.ebi.ac.uk/vol1/ftp/phase3/data/NA12878/alignment/NA12878.mapped.ILLUMINA.bwa.CEU.low_coverage.20121211.bam
- ftp://ftp.1000genomes.ebi.ac.uk/vol1/ftp/phase3/data/NA12878/high_coverage_alignment/NA12878.mapped.ILLUMINA.bwa.CEU.high_coverage_pcr_free.20130906.bam

ATAC-seq and single cell ATAC-seq data for GM12878 generated by Buenrostro et al. were downloaded through GEO with accession number GSE47753 and GSE65360, respectively (*Buenrostro et al., 2013*; *Buenrostro et al., 2015*). Bulk ATAC-seq data from normal donors and AML patients generated by *Corces et al. (2016)*, were downloaded through GEO with accession number GSE74912. Single cell ATAC-seq data for leukemia stem cell and leukemic blasts generated by the same study were downloaded through GEO with accession number GSE74310. Single cell ATAC-seq from normal HSC generated by *Buenrostro et al. (2018)*, were downloaded through GEO with accession number GSE96772. *Supplementary file 1* summarized the detail information of all the datasets used in this study.

## Comparison of mitochondrial genome capture rate and coverage

Sequencing reads from ATAC-seq were aligned to the reference genome by BWA alignment tool (*Li and Durbin, 2009a*). The same reference, GRCh37(used by 1000 genome) and human reference mtDNA sequence rCRS (revised Cambridge reference sequence), were used for ATAC-seq data processing. Samtools (*Li et al., 2009b*) was used for manipulating sequence reads and calculating sequence depth. For all the data sets, the aligned reads were further filtered with mapping quality (Q > 30) and PCR redundancy was removed. The percentage of reads from mitochondrial genome compared to that of the nuclear genome were calculated after all the clean-up steps. The mitochondrial genome coverage was calculated using bases with sufficient sequence quality score (q > 30). A strong depletion region around 3107 due to the sequencing error(3170N) in the reference genome was excluded in the coverage plot (*Ju et al., 2014*).

## Bulk ATAC-seq data process and mitochondrial DNA variants calling

Most of the ATAC-seq pipelines remove mtDNA during their process. To rescue the genetic information from mtDNA, we modified our ATAC-seq pipeline and added SNP calling steps, which focuses on the mitochondrial genome. Briefly, adaptor sequences were trimmed from FASTQs using custom Python scripts. Paired-end reads were aligned to the reference genome using BWA. To improve the accuracy of heteroplasmic mutation calling, we followed the somatic mutation calling guidelines from GATK (*McKenna et al., 2010*), with additional clean-up steps before variant calling. Reads mapped to mtDNA were extracted using Samtools (*Li et al., 2009b*) from the final bam files and variants were called using VarScan2 (*Koboldt et al., 2012*) with '–min-var-freq 0.001' (*Figure 1—figure supplement 1A*). The heteroplasmic variants were further filtered through the following steps to exclude potential sequencing or mapping errors:

1. Thirteen frequent false-positive variants by misalignment due to extensive level of homopolymers in rCRS and due to sequencing error in the reference genome (reported in the previous study [*Ju et al., 2014*]), were also observed and removed in this study. The following sites were explicitly removed:

> Misalignment due to ACCCCCCCTCCCCC (rCRS 302–315)
> A302C, C309T, C311T, C312T, C313T, G316C
> Misalignment due to GCACACACACACC (rCRS 513–525)
> C514A, A515G, A523C, C524G
> Misalignment due to 3107N in rCRS (ACNTT, rCRS 3105–3109)
> C3106A, T3109C, C3110A

2. Strand imbalance is a potential feature of sequencing error with various causes. To remove the potential sequence error from Illumina NextSeq (with a known high error rate at A bases) and sequence error from DNA damage(G->T, C->A) (*Chen et al., 2017*), we required >2 reads detected from both the forward and reverse orientation, and strand is balanced (30% < forward/(forward +reverse)<70%).

3. Variant sites with VAF >0.9, but less than 1, were counted as homoplasmic variants. Although the germline polymorphic can be a backward heteroplasmic mutations, the observation of these events is higher than expected, which implies the false positive calling due to mapping bias for non-reference allele and sequencing errors.

4. For bulk ATAC-seq data from AML patients, heteroplasmic mutations with variant allele frequency >1% were reported.

For all the AML cases (n = 15) from *Corces et al. (2016)*, we selected the cases (n = 12) with at least one confident heteroplasmic mtDNA mutation detected in any cell type for lineage relationship comparison. We found that in one patient (SU209), the number of heteroplasmic mutations (37) and their VAF are significantly higher than other patients. Most of these heteroplasmic mutations also overlapped with common variants present in the general human population (http://ftp.1000genomes.ebi.ac.uk/vol1/ftp/release/20130502/ALL.chrMT.phase3_callmom-v0_4.20130502.genotypes.vcf.gz), which indicates potential sample contamination. Therefore, this case was excluded from lineage relationship comparison and 11 AML cases were finally shown in *Figure 2—figure supplement 1*.

## Single cell ATAC library resequencing

To better evaluate the detection rate in single cell ATAC-seq data, we re-sequenced the previous libraries (LSCs and AML blasts from SU070 and SU373) from *Corces et al. (2016)*. The re-sequenced data were uploaded to GEO and accession number is GSE122576.

## Human AML samples

Human AML samples were obtained from patients at the Stanford Medical Center with informed consent, according to institutional review board (IRB)-approved protocols (Stanford IRB, 18329 and 6453). Mononuclear cells from each sample were isolated by Ficoll separation, resuspended in 90% FBS +10% DMSO, and cryopreserved in liquid nitrogen. All analyses conducted here on AML cells used freshly thawed cells.

## Cell sorting

Cell samples were first thawed and incubated at 37°C with 200 U/mL DNase in IMDM +10% FBS. To enrich for CD34 +cells, magnetic bead separation was performed using MACS beads (Miltenyi Biotech) according to the manufacturer's protocol.

For cell staining and sorting, the following antibody cocktail was used with the schema shown in *Figure 2—figure supplement 3*:

> CD34-APC, clone 581, Biolegend, at 1:50 dilution.
> CD38-PE-Cy7, clone HB7, Biolegend, 1:25 dilution.
> CD19-PE-Cy5, clone H1B9, BD Biosciences, 1:50 dilution
> CD20-PE-Cy5, clone 2H7, BD Biosciences, 1:50 dilution
> CD3-APC-Cy7, clone SK7, BD Biosciences, 1:25 dilution
> CD99-FITC, clone TU12, BD Biosciences, 1:20 dilution
> TIM3-PE, clone 344823, R and D Systems, 1:20 dilution
> CD45-KromeOrange, clone J.33, Beckman Coulter at 1:25 dilution

Samples were sorted using a Becton Dickinson FACS Aria II. pHSCs were re-suspended and kept in cold FACS buffer containing 1 ug/mL propidium iodide prior to and after sorting. Cells were then immediately prepared for single cell ATAC-seq.

## Single cell ATAC-seq from pHSC

Cells were washed two times in C1 DNA Seq Cell Wash Buffer (Fluidigm).~10 K cells were then re-suspended in 6 mL of C1 DNA Seq Cell Wash Buffer, and were combined with 4 mL of C1 Cell Suspension Reagent, 7 mL of this cell mix was loaded onto the Fluidigm IFC. Cells at a concentration of 260–380 cells/μL were then assayed using scATAC-seq as previously described (*Buenrostro et al., 2015*). Briefly, single cells were captured using the C1 Single-Cell Auto Prep IFC microfluidic chips. Cells were permeabilized and accessible fragments were captured using 20 μL of Tn5 transposition mix (1.5x TD buffer, 1.5 μL transposease (Nextera DNA Sample Prep Kit, Illumina), 1x C1 Loading Reagent with low salt (Fluidigm), and 0.15% NP40) at 30 min at 37°C. In a 96-well plate, 7 μL of harvested libraries were amplified in 50 μL PCR for an additional 17 cycles (1.25 μM custom Nextera dual-index PCR primers in 1x NEBnext High-Fidelity PCR Master Mix using the following PCR conditions: 72°C for 5 min; 98°C for 30 s;) using the following PCR conditions: 72°C for 5 min; 98°C for 30 s; and thermocycling at 98°C for 10 s, 72°C for 30 s, and 72°C for 1 min. The PCR products were pooled creating a final volume of ~4.8 mL. The pooled library was purified on a single MinElute PCR purification column (Qiagen). Libraries were quantified using qPCR prior to sequencing. The scATAC-seq libraries were sequenced by Illumina MiSeq. The sequence data were uploaded to GEO under the accession number GSE122577.

## Single cell ATAC-seq data processing and mitochondrial DNA variant calling

Single cell ATAC-seq were processed similarly to the bulk ATAC-seq, taking each individual cell as one sample. Recalibration steps were not applied for single cell data, as the sequence depth is not sufficient to empirically adjust the quality scores. After cleaning the alignment, files from every single cell were merged and heteroplasmic variants were first called with the merged bam and filtered using the same criteria as bulk data. Heteroplasmic variants called from merged data or from bulk

data were re-counted in each individual cell using Samtools with '-q 20 -Q 20'. And the non-reference allele had to match the variants detected in merged or bulk data.

## Detection rate estimation

In every single cell, if the variant allele detected in merged or bulk data were supported by any reads, it was considered positive; otherwise, it was counted as zero. A binary matrix was used to present the lineage relationship among single cells and plotted as a heat map. The intersections of the variants were quantified by the Upset R package (*Conway et al., 2017*). The number of detected variants showed a correlation with sequencing depth and the number of cells with all variants (*Figure 1—figure supplements 3* and *4*) confirmed the variants already co-existed in the ancestral cell. Following this assumption, the detection rate can be measured as the proportion of cells with variants in the total number of cells. For each variant, cells were separated into different bins, increased by 10, according to the total sequencing depth at each variant. The detection rate for each variant site was then calculated in each bin. The combined detection rate was estimated by $1-(1\ R_1)*(1\ R_2)*(1\ R_3)*(1\ R_4)$, where $R_n$ is the detection rate for each variant.

## Lineage inference

The probability of observing a mutation at a given site is $P_n = n*r$, where $r$ is the average mutation rate in the mitochondrial genome and $n$ is the copy of mtDNAs in a single cell. $r$ is estimated to be $\sim 10^{-7}$ per base (*Coller et al., 2001*), n is around 100 ~ 10000 per cell (*Miller et al., 2003*), so $P_n$ will be $10^{-5} \sim 10^{-3}$. The probability of N cells sharing the same mtDNA mutations, but arising independently, will be $(P_n)^N$. Thus, when there are more than 3 cells in the population sharing a common mtDNA mutation, the probability of these independently occurring will be close to 0. Cells with common mtDNA mutations having inherited the mutations from the same ancestral cell is more likely to explain the observed result. Furthermore, when a set of mutations (more than 1) is detected in more than 1 cells, the null hypothesis (independent occurrence) is rejected more confidently. The mutations within the ancestral cells can be inferred from the intersection of mutations. If a set of mutations co-existed in the ancestral cell, the absence of one of several linked mutations in the daughter cells is more likely caused by false negative detection in single-cell libraries or genetic draft during cell replication. Then the observed cells with different intersections (e.g *V1 +V2*) are expected to be given by $P_{v1}*P_{v2}*N$, after normalization by sequencing depth. The divergence of intersections from high-frequency mutations indicates the separation of mtDNA mutations and multiple cell lineage. The intersections of the variants were quantified by the Upset R package (*Conway et al., 2017*). In the scATAC-seq from pHSCs from SU353, the intersection of variants showed most of the cells were separated by two sets of different variants (*Figure 2D*). But there are a few cells displaying a mixture of variants from the two sets. We suspected these may cause by the doublet of cells in the same well during single cell separation on C1 chip. We further separated the intersection map by the chip and observed the number of cells with mixture variants correlated to the concentration of cells loaded to C1 Chip. These cells were removed during subsequent analysis. Single cells with any variants in the two sets were kept and cells with more than 40X coverage of mtDNA, but with no variants in the two sets were considered as HSCs with WT mtDNA. After all the filter steps, 153 pHSC cells had lineage information and were separated into three subgroups.

## Single cell ATAC-seq chromatin analysis

ATAC sequences mapped to the nuclear genome were used for chromatin accessibility profiling. Bam files were merged for the same cell types and used as input files for chromVAR (*Schep et al., 2017*). Peak files from *Buenrostro et al. (2018)* were used as open background regions to quantify the accessibility signal from every single cell. Cells with fewer than 200 unique reads or less than 25% of reads in peak regions were removed for chromatin analysis. chromVAR was applied to calculate TF motif-associated chromatin accessibility landscape changes and identify potential regulators of epigenomic variability. This approach quantifies accessibility variation across single-cells by aggregating accessible regions containing a specific TF motif, then compares the observed accessibility of all peaks containing a TF motif to a background set of peaks normalizing for known technical confounders. For determining differentially accessible motifs between different subpopulations, a Wilcoxon test was used to calculate the p values of the difference between the two groups.

## Code availability

Custom analysis code can be downloaded from GitHub (https://github.com/ChangLab/ATAC_mito_sc; *Xu, 2019*; copy archived at https://github.com/elifesciences-publications/ATAC_mito_sc).

## Acknowledgements

We thank Christina Curtis, Ava Carter, Furqan Fazal, Kevin Parker and Chun-Kan Chen for insightful advice and assistance. Supported by US National Institutes of Health P50-HG007735 (to HYC), R01CA188055 (to RM), and R01HL142637 (to RM). RM is a Scholar of the Leukemia and Lymphoma Society. HYC is an Investigator of the Howard Hughes Medical Institute.

## Additional information

### Competing interests

Ravindra Majeti: is a co-founder, consultant, equity holder, and serves on the Board of Directors of Forty Seven Inc. Ravindra Majeti is Reviewing editor, *eLife*. Howard Y Chang: is a co-founder of Accent Therapeutics and an advisor for 10x Genomics and Spring Discovery. Stanford University has filed a patent on ATAC-seq(US20160060691A1), on which HYC is named as an inventor. The other authors declare that no competing interests exist.

### Funding

| Funder | Grant reference number | Author |
| --- | --- | --- |
| National Human Genome Research Institute | P50-HG007735 | Howard Y Chang |
| Howard Hughes Medical Institute | | Howard Y Chang |
| National Cancer Institute | R01HL142637 | Ravindra Majeti |
| National Cancer Institute | R01CA188055 | Ravindra Majeti |

The funders had no role in study design, data collection and interpretation, or the decision to submit the work for publication.

### Author contributions

Jin Xu, Conceptualization, Data curation, Software, Formal analysis, Investigation, Methodology, Writing—original draft, Writing—review and editing; Kevin Nuno, Resources, Data curation, Investigation, Methodology, Writing—review and editing; Ulrike M Litzenburger, Resources, Data curation, Investigation, Methodology; Yanyan Qi, Resources, Data curation, Investigation; M Ryan Corces, Resources, Data curation; Ravindra Majeti, Resources, Data curation, Supervision, Funding acquisition, Methodology, Writing—review and editing; Howard Y Chang, Conceptualization, Supervision, Funding acquisition, Writing—original draft, Writing—review and editing

### Author ORCIDs

Jin Xu (iD) http://orcid.org/0000-0003-0944-9835
Howard Y Chang (iD) http://orcid.org/0000-0002-9459-4393

### Ethics

Human subjects: AML samples were obtained from patients at the Stanford Medical Center with informed consent, according to institutional review board (IRB)-approved protocols (Stanford IRB, 18329 and 6453).

### Decision letter and Author response

Decision letter https://doi.org/10.7554/eLife.45105.032
Author response https://doi.org/10.7554/eLife.45105.033

# Additional files

## Supplementary files

• Supplementary file 1. Information of datasets utilized in this study.
DOI: https://doi.org/10.7554/eLife.45105.013

• Supplementary file 2. Heteroplasmic mtDNA mutations detected in each AML patient. Allele frequence, seqeunce coverage and annotation information of the mtDNA variants are provided.
DOI: https://doi.org/10.7554/eLife.45105.014

• Supplementary file 3. Relative merits of mtDNA vs. other genetic markers for lineage tracing.
DOI: https://doi.org/10.7554/eLife.45105.015

• Transparent reporting form
DOI: https://doi.org/10.7554/eLife.45105.016

## Data availability

Sequencing data have been deposited in GEO under accession codes GSE122576 and GSE122577.

The following datasets were generated:

| Author(s) | Year | Dataset title | Dataset URL | Database and Identifier |
|---|---|---|---|---|
| Xu J | 2018 | Single cell lineage tracing by endogenous mitochondrial DNA mutations in ATAC-seq data | https://www.ncbi.nlm.nih.gov/geo/query/acc.cgi?acc=GSE122576 | NCBI Gene Expression Omnibus, GSE122576 |
| Xu J, Chang HY | 2018 | Single cell lineage tracing by endogenous mitochondrial DNA mutations in ATAC-seq data | https://www.ncbi.nlm.nih.gov/geo/query/acc.cgi?acc=GSE122577 | NCBI Gene Expression Omnibus, GSE122577 |

The following previously published datasets were used:

| Author(s) | Year | Dataset title | Dataset URL | Database and Identifier |
|---|---|---|---|---|
| Buenrostro JD, Giresi PG, Zaba LC, Chang HY, Greenleaf WJ | 2013 | Transposition of native chromatin for fast and sensitive epigenomic profiling of open chromatin, DNA-binding proteins and nucleosome position | https://www.ncbi.nlm.nih.gov/geo/query/acc.cgi?acc=GSE47753 | NCBI Gene Expression Omnibus, GSE47753 |
| Buenrostro JD | 2015 | Single-cell chromatin accessibility data using scATAC-seq | https://www.ncbi.nlm.nih.gov/geo/query/acc.cgi?acc=GSE65360 | NCBI Gene Expression Omnibus, GSE65360 |
| Buenrostro JD | 2016 | ATAC-seq data | https://www.ncbi.nlm.nih.gov/geo/query/acc.cgi?acc=GSE74912 | NCBI Gene Expression Omnibus, GSE74912 |
| Buenrostro JD | 2016 | Single-cell chromatin accessibility data using scATAC-seq | https://www.ncbi.nlm.nih.gov/geo/query/acc.cgi?acc=GSE74310 | NCBI Gene Expression Omnibus, GSE74310 |
| Buenrostro JD | 2018 | Single-cell epigenomics maps the continuous regulatory landscape of human hematopoietic differentiation | https://www.ncbi.nlm.nih.gov/geo/query/acc.cgi?acc=GSE96772 | NCBI Gene Expression Omnibus, GSE96772 |

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
