## [Decision Letter]

Thank you for submitting your article "Single-cell lineage tracing by endogenous mutations enriched in transposase accessible mitochondrial DNA" for consideration by *eLife*. Your article has been reviewed by three peer reviewers, including Ross L Levine as the Reviewing Editor and Reviewer #1, and the evaluation has been overseen by Marianne Bronner as the Senior Editor.

The reviewers have discussed the reviews with one another and the Reviewing Editor has drafted this decision to help you prepare a revised submission.

Summary:

The work was thought to be of significant interest, particularly as a novel approach to lineage tracing. The reviewers did identify some areas where further work and clarification would strengthen the paper.

Essential revisions:

1) One important feature of mtDNA sequencing the authors mention is the possibility of horizontal transfer of mitochondrial DNA. This is especially interesting in the context of mtDNA mutations found at low VAF across many single cells. Can the authors determine if such low level of VAF is above the rate of horizontal transfer, or even the experimental doublet rate? A mixing experiment between mouse and human cells should solve this quickly.

2) The authors demonstrate that ATAC-seq enriches for mtDNA compared to whole exome sequencing and whole genome sequencing (Figure 1B, 1C and Figure 1—figure supplement 2). However, there is an important number of heteroplasmic mtDNA mutations (18) which are detected by high-coverage whole genome sequencing (WGS_H), which have failed to be detected by ATAC-seq (Figure 1—figure supplement 2B). How can this be explained, given the fact that ATAC-seq leads to higher coverage of mitochondrial genome in respect to other approaches? Would a higher number of ATAC-seq reads lead to the identification of more existing mtDNA mutations? Do these 18 mtDNA mutations reside in a specific region of the mitochondrial genome? I would like to ask the authors to perform the same analysis and compare the mtDNA coverage and the number of mutations after ATAC-seq and WGS-H in one more human cell line, in order to validate their results from GM12878 cells.

---

## [Author Response]

Essential revisions:1) One important feature of mtDNA sequencing the authors mention is the possibility of horizontal transfer of mitochondrial DNA. This is especially interesting in the context of mtDNA mutations found at low VAF across many single cells. Can the authors determine if such low level of VAF is above the rate of horizontal transfer, or even the experimental doublet rate? A mixing experiment between mouse and human cells should solve this quickly.

We investigated the possibility of horizontal mitochondrial transfer by examining the single cell ATAC-seq dataset from a mixing experiment with human and mouse cells (Satpathy et al., 2018), as suggested by the reviewer. We have added this analysis to the revised manuscript (Figure 2—figure supplement 4). The human and mouse cells were co-cultured for 6 hours prior to the single cell analysis. We mapped the mtDNA reads to the human and mouse references separately. As shown in Figure 2—figure supplement 4A-B, we found that mtDNA strictly tracked with species-of-origin along with the genomic DNA, with the exception of one pair of cell doublets (in red) that was detected on both genomic and mtDNA. Species-specific mtDNA always paired with species-specific nuclear genomic DNA (Figure 2—figure supplement 4C). These results suggest that mitochondrial horizontal transfer is not a confounder of our study.

Furthermore, the literature reports that the rare events of mitochondrial DNA transfer is unidirectional from stromal cells to AML cells, and requiring cell-to-cell contact. There is an average increasing of 14% in the mitochondrial mass of the receiver cells, which is ~17 copies of mtDNA (Moschoi et al., 2016). With this knowledge, we would expect that: 1) Mitochondria donors are normal cells, which would not accumulate high frequency mtDNA mutations. Transfer of wild type mtDNA would have no impact on lineage tracing. 2) The variant allele frequency from the donor cell will be diluted by ~8 fold in the receiving cells, even assuming the extremely unlikely scenario of selective transfer of mutant mtDNA into neighbor cells. Thus, we would expect the variant allele frequency from horizontal transfer to be much lower than the endogenous variants, and not substantially impact the results.

2) The authors demonstrate that ATAC-seq enriches for mtDNA compared to whole exome sequencing and whole genome sequencing (Figure 1B, 1C and Figure 1—figure supplement 2). However, there is an important number of heteroplasmic mtDNA mutations (18) which are detected by high-coverage whole genome sequencing (WGS_H), which have failed to be detected by ATAC-seq (Figure 1—figure supplement 2B). How can this be explained, given the fact that ATAC-seq leads to higher coverage of mitochondrial genome in respect to other approaches? Would a higher number of ATAC-seq reads lead to the identification of more existing mtDNA mutations? Do these 18 mtDNA mutations reside in a specific region of the mitochondrial genome?

In Figure1—figure supplement 2B, we showed there are 18 heteroplasmic mtDNA mutations that were uniquely detected by high coverage whole genome seqeuencing (WGS_H), and 11 heteroplasmic mtDNA mutation uniquely detected by ATAC-seq in cultured human GM12878 lymphoblastoid cells. These method-specific mutations have VAF lower than 0.5% (Figure 1—figure supplement 3C). These low frequency mutations can only be detected when the sequence coverage is very high (~10,000X), and are not strong signals that drive the lineage tracing results because they are very infrequently detected. These method-specific, low frequency mutations are scattered across mtDNA. As requested, we added a new panel(C) to Figure 1—figure supplement 2 to provide the allele frequency and the position of these heteroplasmic mtDNA mutations detected by WGS_H and ATAC-seq.

We speculate that the low frequency mtDNA mutations may be new mutations that arose during serial passage in cell culture. The published WGS_H and ATAC-seq data of GM12878 cells were not concurrently obtained. All of the WGS_H and the ATAC specific mutations showed a strong enrichment of C->T mutations, which match the mitochondrial specific mutation spectrum and are consistent with our hypothesis.

I would like to ask the authors to perform the same analysis and compare the mtDNA coverage and the number of mutations after ATAC-seq and WGS-H in one more human cell line, in order to validate their results from GM12878 cells.

We analyzed human K562 cells, a myeloid leukemia cell line with both whole genome sequencing and ATAC-seq data available. These results have been added to the revised manuscript (Figure 1—figure supplement 3). We observed a 35-fold enrichment of mtDNA sequence in ATAC-seq data compared to whole genome sequencing (Figure 1—figure supplement 3A). The average coverage of the mitochondrial genome is 16,928 and 5,940 for WGS and ATAC-seq, respectively (Figure 1—figure supplement 3B). We detected 87 heteroplasmic mtDNA mutations in total with the same criteria as described in the Materials and methods section. 36 of these mutations can be detected by both WGS and ATAC, while 51 of them are only detected in either WGS or ATAC (Figure 1—figure supplement 3C). Similarly, these method-specific heteroplasmic mtDNA mutations have VAF lower than 0.5% as observed in GM12878 (Figure 1—figure supplement 3D).